# Genome-Wide Characterization of High-Affinity Nitrate Transporter 2 (NRT2) Gene Family in *Brassica napus*

**DOI:** 10.3390/ijms23094965

**Published:** 2022-04-29

**Authors:** Run-Jie Du, Ze-Xuan Wu, Zhao-Xi Yu, Peng-Feng Li, Jian-Yu Mu, Jie Zhou, Jia-Na Li, Hai Du

**Affiliations:** 1College of Agronomy and Biotechnology, Southwest University, Chongqing 400716, China; dddrj12345@email.swu.edu.cn (R.-J.D.); w1045123799@126.com (Z.-X.W.); yzx20010120@gmai.com (Z.-X.Y.); pengfengli17@126.com (P.-F.L.); mujianyu728@126.com (J.-Y.M.); zj893422105@163.com (J.Z.); ljn1950@swu.edu.cn (J.-N.L.); 2Academy of Agricultural Sciences, Southwest University, Chongqing 400716, China

**Keywords:** *Brassica napus*, NRT2, gene family, evolution, nitrate, expression profile

## Abstract

Nitrate transporter 2 (NRT2) plays an essential role in Nitrogen (N) uptake, transport, utilization, and stress resistance. In this study, the NRT2 gene family in two sequenced *Brassica napus* ecotypes were identified, including 31 genes in ‘Zhongshuang11’ (*BnaZSNRT2s*) and 19 in ‘Darmor-*bzh*’ (*BnaDarNRT2s*). The candidate genes were divided into three groups (Group I−III) based on phylogenetic analyses, supported by a conserved intron-exon structure in each group. Collinearity analysis revealed that the large expansion of *BnaZSNRT2s* attributed to allopolyploidization of ancestors *Brassica rapa* and *Brassica oleracea*, and small-scale duplication events in *B. napus*. Transcription factor (TF) binding site prediction, *cis*-element analysis, and microRNA prediction suggested that the expressions of *BnaZSNRT2s* are regulated by multiple factors, and the regulatory pattern is relatively conserved in each group and is tightly connected between groups. Expression assay showed the diverse and differentiated spatial-temporal expression profiles of *BnaZSNRT2s* in Group I, but conserved patterns were observed in Group II/III; and the low nitrogen (LN) stress up-regulated expression profiles were presented in Group I−III, based on RNA-seq data. RT-qPCR analyses confirmed that *BnaZSNRT2.5A-1* and *BnaZSNRT2.5C-1* in Group II were highly up-regulated under LN stress in *B. napus* roots. Our results offer valid information and candidates for further functional *BnaZSNRT2s* studies.

## 1. Introduction

Nitrogen (N), mainly in the form of nitrate, is a crucial element for plant development and a vital factor affecting diverse plant bioprocesses, such as photosynthesis and protein synthesis [1,2]. Thus, it is substantially required by crops to form agricultural production [3,4,5,6]. Normally, sufficient N conditions can protect crop yield and quality in variable environments [7]. However, the distributions of biologically available N, including nitrate, ammonium, and small peptides, are uneven in natural and agricultural land worldwide [8]. In this case, large amounts of fertilizer were generally applied in production, and approximately 60% of the annual fertilizer consumption is N, which ensures more than 40% of the population’s basic food needs [6,9]. On the other hand, excessive N fertilizer application brings a series of environmental damages, such as acid rain [10], water eutrophication [11,12], the greenhouse effect [13,14], and poor soil fertility [15]. Hence, analyses concerning increasing the efficiency in N absorption and utilization perform crucial roles in crop yield, quality, and even environmental protection.

NITRATE TRANSPORTER 2 (NRT2) homologous proteins are typical high-affinity nitrate transporters in the plant kingdom and are responsible for the nitrate uptake process in plants. In general, NRT2 proteins (NRT2s) have a typical membrane topology connected by a cytosolic loop, including 1 MFS domain that exhibits dual affinities for nitrate [16] and 12 transmembrane domains [17], which are usually located on the cell plasma membrane [18]. In *Arabidopsis*, there are 7 NRT2 genes (*AtNRT2s*) that were divided into three groups [6,19]; with Group I containing 5 members while Groups II/III only include 1 member each. In Group I, *AtNRT2.1* and *AtNRT2.2* play vital roles in nitrate uptake in roots [20]; *AtNRT2.4* acts a key role in nitrate uptake under low nitrogen (LN) stress in both shoots and roots [21]; *AtNRT2.6* is involved in biotic and abiotic stresses [22]; whereas *AtNRT2.3* remains to be functional elucidated yet. In Group II, *AtNRT2.5* plays an essential role in severe nitrogen starvation response. The expression of *AtNRT2.5* was highly induced after a long N starvation, and then it acted as the major transporter for high-affinity nitrate uptake [23]. In Group III, *AtNRT2.7* is expressed in tonoplast and contributes to the N accumulation in seeds [24]. Notably, except for *AtNRT2.7*, all of the other AtNRT2 transporters could interact with *AtNAR2.1,* enhancing the nitrate uptake capacity of *AtNRT2s* [25]. Similarly, the homologs of *AtNRT2s* in other plants were widely demonstrated to perform numerous roles in N uptake, transport, and utilization processes across developmental stages. For instance, in rice, *OsNRT2.3a* plays a key role in increasing N use efficiency and field [26]. In wheat, *TaNRT2.5* is expressed in the embryo and shell and acts in nitrate accumulation in seeds [27]. In maize, *ZmNRT2.1* regulates nitrate uptake along the root axis [28]. Together, *NRT2* homologs play key roles in nitrate uptake and even utilization in plants. Thus, systematically identifying the NRT2 gene families in plant genomes and exploring their roles involved in nitrate utilization processes may contribute to promoting the N utilization efficiency (NUE) and crop yields without resorting to excessive N fertilizer.

*Brassica napus* (*B. napus*; A_n_A_n_C_n_C_n_, 2n = 38) is a significant oil crop worldwide which was an allopolyploid produced by *Brassica rapa* (*B. rapa*; A_n_A_n_, n = 10) and *Brassica oleracea* (*B. oleracea*; C_n_C_n_, n = 9) ~7500 years ago [29,30]. So far, the genomes of 2 *B. napus* ecotypes have been sequenced, namely ‘Darmor-*bzh*’ and ‘Zhongshuang 11’ (ZS11). *B. napus* rely on the amount of N in the production, which needs more N fertilizer to produce a unit of yield than wheat and rice [31,32]. However, the NUE of *B. napus* is much lower than wheat and rice, causing a mass of N loss [33,34]. Given the important roles in N utilization-related processes, identifying the NRT2 encoding genes at a genome-wide level and exploring their roles in N uptake and utilization processes in *B. napus* has potential research significance and application value, aiming to improve the NUE and even the yield and quality of *B. napus*.

In our study, we conducted global analyses of the NRT2 gene family in 2 ecotypes of *B. napus* genomes (‘Darmor-*bzh*’ and ‘ZS11’) at the genome-wide level, accompanied by a series of bioinformatics assays of the candidates, including sequence structure, phylogenetic relationship, chromosomal location, collinearity relationship, gene duplication, regulatory mechanism prediction, etc. Then, we analyzed the spatial-temporal expression profiles of candidates in 52 *B. napus* (ZS11) tissues/organs across distinct developmental stages. Moreover, the LN stress expression patterns of the candidates in *B. napus* (ZS11) seedling roots were analyzed by RNA-Seq and RT-qPCR methods, respectively. Our findings constitute the first step toward further research on the molecular functions of *NRT2s* in *B. napus*.

## 2. Results

### 2.1. Identification of NRT2 Genes in B. napus

To identify the NRT2s in *B. napus* genome, we performed preparatory BLASTP and Tblastn searches using the protein sequences of *Arabidopsis* NRT2 protein (AtNRT2s) [35] as queries. Two sequenced genome databases of *B. napus* varieties in GENOSCOPE (Darmor–*bzh*, http://www.genoscope.cns.fr/brassicanapus/, accessed on 19 August 2014) [30] and BnPIR (Zhongshuang 11, ZS11, http://cbi.hzau.edu.cn/bnapus/, accessed on 10 September 2020) [29] were used. After excluding the redundant sequences, the remainders were further verified by SMART (http://smart.embl-heidelberg.de/, accessed on 26 October 2020) and ExPASy (https://web.expasy.org/compute_pi/, accessed on 23 February 2022) online software to ensure the candidates contain the typical sequence features of this gene family. Finally, we obtained 19 candidate genes from the ‘Darmor-*bzh*’ genome (*BnaDarNRT2s*) and 31 candidates from the ‘ZS11’ genome (*BnaZSNRT2s*) with relative complete functional domains. Then, we named them based on the *Arabidopsis* homologous *NRT2s* and their chromosome locations in *B. napus*, such as the four homologs of *AtNRT2.1* gene in A_n_ subgenome were named as *BnaZSNRT2.1A-1* to *BnaZSNRT2.1A-4* whereas these in C_n_ were named as *BnaZSNRT2.1C-1* to *BnaZSNRT2.1C-4* (Table 1 and Appendix A).

As shown in Table 1 and Appendix A, the length of BnaDarNRT2 proteins (BnaDarNRT2s) and BnaZSNRT2 proteins (BnaZSNRT2s) ranged from 154 to 506 amino acids and 113 to 567 amino acids. The average value was 427.16 and 472.26, and the standard deviation (SD) was 99.54 and 126.73, respectively. Isoelectric point (pI) of BnaDarNRT2s and BnaZSNRT2s varied from 7.54 to 9.93 and 6.02 to 9.93, with an average value of 8.94 and 8.65, and the SD was 0.57 and 0.87, respectively. Their molecular weight (MW) varied from 17.09 to 54.83 kDa and 12.19 to 61.87 kDa, with the average value being 46.44 and 51.25, and the SD 10.7 and 13.75, respectively. Subcellular localization analysis showed that nearly all 19 BnaDarNRT2s and 30 BnaZSNRT2s were located on the cell membrane. Only BnaZSNRT2.6C-5 was located on chloroplast/cytoplasm, which suggested their potential function features in the nitrate utilization process.

### 2.2. Phylogenetic and Sequence Structure Analysis of B. napus NRT2 Gene Family

To investigate the phylogenetic relationship of the candidate *NRT2s*, a neighbor-joining (NJ) phylogenetic tree of the 19 BnaDarNRT2s, 31 BnaZSNRT2s, and 7 AtNRT2s was generated based on the multi-alignment of the whole-length protein sequences (Figure 1A). Due to technical reasons (having no common information site(s) between sequences), *BnaZSNRT2.8C-1* was excluded from the phylogenetic tree because of severe sequence deletion. According to the topology and bootstrap values of the NJ tree, the candidates were separated into three groups: Group I–III. Group I was the largest, which contained 15 *BnaDarNRT2s*, 26 *BnaZSNRT2s,* and 5 *AtNRT2s*; Group II/III both contained 2 *BnaDarNRT2s*, 2 *BnaZSNRT2s,* and 1 *AtNRT2s*. The number of *BnaDarNRT2s* and *BnaZSNRT2s* in Group II and III was equal, while the number of homologs between these two ecotypes was quite different in Group I. Excepting for 11 homologous gene pairs, there are 4 *BnaDarNRT2s* and 15 *BnaZSNRT2s* having non-homologs in Group I. In general, the *AtNRT2s* have homologs in both ecotypes, and the *AtNRT2s* in Group I have more homologs in ‘ZS11’ than in ‘Darmor-*bzh*’, excepting *AtNRT2*.6, which only has *BnaZSNRT2s* homologs. Subsequently, we analyzed the sequence identity and similarity of the full-length DNA, CDS, and protein sequences of each homologous gene pair (Appendix A). The results showed that the sequence identity and similarity in Group II/III were very high, with the protein sequence identity ranging from 84.6–100% in each group, while that was decreased in Group I (ranging from 62.10–100%). These results indicated that the gene number, sequence features, and even functions of the homologous gene pairs in Group II and III were highly conserved during evolution, while those in Group I may undergo functional diversification.

Sequence structure feature observed 8 relatively conserved intron insertion sites in terms of conserved insertion site and phase, namely intron “1” to “8” (Figure 1C). The intron number of *NRT2s* varies in Group I−III or even within a group, while members of Group I had 1 to 5 introns, Group II had 2 introns, and Group III had 1 intron. Although the number of introns was diverse in the three groups, the intron insertion site and phase were generally conserved within each group. Members of Group II contained intron “4” and “7”, which were conserved in this group. Similarly, the intron patterns were also completely conserved in group III, members of which contained intron “7”. By contrast, the intron number, insertion site, and phase in Group I were diverse and relatively less conserved, which contained 6 relative conserved intron sites, “1–3”, “5”, “6”, and “8”. Among them, intron “5” (39/46, ~85%) and “6” (37/46, ~80%) were highly conserved in this group, with several aberrations due merely to the severe sequence missing and flanking sequence diversity. In contrast, the rest introns (“1”, “2”, “3”, and “8”) were only conserved within several genes. Together, the exon-intron structures of Group II and III were highly conserved in comparison with that of Group I. The intron “7” was shared and completely conserved in Group II and III, implying the close relationship between these two groups. Moreover, almost all introns were located outside of the transmembrane domains (TMs), except introns “5” and “8” in Group I.

The protein domain prediction using HMMER online software (https://www.ebi.ac.uk/Tools/hmmer/, accessed on 19 November 2021) showed that 42 of the 50 members (84%) of BnaDarNRT2s and BnaZSNRT2s have 8 to 12 TMs (Figure 1B). Most members of Group I (34/41, ~82.93%) had 8 to 12 TMs, all members of Group II possessed 11 TMs, and nearly all of Group III had 12 TMs except AtNRT2.7 (10 TMs). The MFS-1 domain existed in most candidates (Figure 1C), which was located in the middle region, covering nearly all TMs. Moreover, the distribution and characteristics of the MFS-1 and TM domains were highly conserved within each group, especially in Groups II and III (Appendix A). Our results demonstrated that the protein domains were relatively conserved in the NRT2 gene family, and they were highly conserved in the same group.

### 2.3. Chromosomal Location and Collinearity Relationship in BnaZSNRT2s

As shown in Figure 2A, the 31 *BnaZSNRT2s* were scattered on 12 of the 19 *B. napus* chromosomes. There are 5, 4, and 4 *BnaZSNRT2s* located on chromosomes C_n_08, A_n_06, and C_n_03, respectively; each of the last 9 chromosomes contained 2 *BnaZSNRT2s*. The numbers of *BnaZSNRT2s* on A_n_-subgenome (14 genes) and C_n_-subgenome (17 genes) showed a biased trend, with more genes on C_n_-subgenome.

The number of *NRT2s* in *B. napus* genome is larger than in the other species reported, e.g., *Arabidopsis* (7 genes) [35], poplar (6 genes) [36], barley (10 genes) [37]. This may because *B. napus* (A_n_A_n_C_n_C_n_, 2n = 38) was newly originated from the hybridization event between *B**rassica rapa* (A_n_A_n_, n = 10) and *B**rassica oleracea* (C_n_C_n_, n = 9) ~7500 years ago [30], and Brassicaceae species underwent a whole-genome triplication (WGT) event [38]. Therefore, in theory, *B. rapa*, *B. oleracea,* and *B. napus* genomes may have 21, 21, and 42 *NRT2s* expanded from the 7 *AtNRT2s*. In fact, only 14, 14, 19, and 31 genes were identified in *B. rapa*, *B. oleracea*, ‘Darmor-*bzh’,* and ‘ZS11’, respectively (Appendix A), indicating that many *NRT2s* may have been lost during evolution. Theoretically, 5 *AtNRT2s* in Group I should expand to 30 *BnaZSNRT2s*/*BnaDarNRT2s*, and 1 *AtNRT2* in Group II/III should expand to 6 homologs in *B. napus,* respectively. In fact, 27, 2, and 2 genes were identified in Group I−III in the ‘ZS11’ecotype, and 15, 2, and 2 genes were identified in Group I−III in the ‘Darmor-*bzh*’ ecotype, indicating that ‘ZS11’ retained 12 genes more than ‘Darmor-*bzh*’ in Group I, and the two ecotypes both lost 4 genes in Group II/III.

Collinearity relationship analysis found that 29 of the 31 (~93.55%) *BnaZSNRT2s* had the collinearity relationship in *B. napus*, *B. rapa*, and/or *B. oleracea*, except for *BnaZSNRT2.6C-2* and *BnaZSNRT2.6C-5* (Figure 2B, Appendix A). Among the 29 *BnaZSNRT2s*, 13 genes (~44.83%) were inherited from allopolyploidy between *B**. rapa* and *B**. oleracea*, including 7 genes (~24.14%) were inherited from *B. rapa* and 6 genes (~20.69%) were from *B. oleracea*; the last 16 (~55.17%) *BnaZSNRT2s* were originated from other duplication events within *B. napus* genome, including 11 (68.75%) genes from segmental duplication (SD), 3 (18.75%) genes from the segmental exchange (SE), and 2 (12.5%) genes from homologous exchange (HE) events. Only 1 tandem duplication (TD) event was identified (*BnaZSNRT2.1C-1*/*BnaZSNRT2.1C-2*). Moreover, all the 3 genes from the SE event were from A_n_-subgenome, which replaced the genes on C_n_-subgenome in ‘ZS11’. This demonstrated bias retention for genes derived from *B. rapa* in *B. napus* after allopolyploidy. Furthermore, 9 (69.23%) of the 13 *BnaZSNRT2s* derived from allopolyploidy have undergone small-scale duplications in the *B. napus* genome as well, including 2 genes that experienced two SD events, 5 genes underwent one SD event, and the other 2 genes underwent one HE events, implying that the larger number of *BnaZSNRT2s* expansion might mainly attribute to allopolyploidy and subsequent SD events in *B. napus*. Additionally, the sequence similarity and identity of the full-length DNA, CDS, and protein sequences of the 8 duplicated gene pairs in *B. napus* were very high, and the sequence identities were on average ~85.74%, ~91.49%, and ~94.25%, respectively (Appendix A). This indicated that the duplicated genes are functionally redundant.

Taken together, our results proved that ‘ZS11’ showed a higher retention rate than ‘Darmor-*bzh*’; allopolyploidy and SD events (24/29, ~83%) mainly contributed to the massive expansion of *BnaZSNRT2s* with these derived from *B. rapa* were inclined to be reserved in *B. napus* genome. In the following sections, we will focus on the features of the candidate genes in the native variety ZS11 ecotype.

### 2.4. Potential Regulatory Mechanism in the Promoter Regions of BnaZSNRT2s

The transcription factor (TF) binding sites in the promoter sequences (−1500bp) of the 31 *BnaZSNRT2s* were predicted by the online PlantTFDB software, and then a regulatory network was generated (Figure 3A). A total of 292 TF binding sites were predicted among 29 *BnaZSNRT2s*, excepting *BnaZSNRT2.7A-1* and *BnaZSNRT2.1C-2* (Appendix A), which belonged to 21 TF families, with Dof (50 sites), bZIP (36 sites), MYB (29 sites) and B3 (22 sites) families contained most sites (Figure 3B). In Group I, the candidates belong to 21 TF families that might regulate 26 *BnaZSNRT2s* with 5 TF families may only regulate 1 gene respectively (e.g., G2like, GRAS, SRS), whereas the last 16 TF families (e.g., Dof, MYB, B3) regulate multi-genes respectively. In Group II, B3, Dof, and C2H2 families might regulate *BnaZSNRT2.5C-1*; SBP and Dof might regulate *BnaZSNRT2.5A-1*. In Group III, both Dof and AP2 might regulate *BnaZSNRT2.7C-1*. The complicated regulatory network by TF in Group I might indicate the diverse expression profiles of *BnaZSNRT2s*.

To understand the potential regulatory mechanism, we subsequently predicted *cis*-acting elements on the −1500bp upstream promoter regions of the 31 *BnaZSNRT2s* by PlantCARE online software. A total of 2941 *cis*-acting elements were found, classified into 56 types (Appendix A). Except for the common basic core elements (e.g., A-box, TATA-box) and light-responsive elements (e.g., ACE), the rest were divided into five major groups, including hormone response element, protein binding site, special response element, tissue-specific expression element, and biological and abiotic response elements (Figure 3C). In the hormone response element group, 12 types of *cis*-elements were predicted, and 35, 24, and 19 *BnaZSNRT2s* might be involved in MeJA (TGACG-motif, CGTCA-motif), abscisic acid (ABRE), and ethylene (ERE) responsive processes. In the protein binding site group, all types of *cis*-acting elements were related to MYB binding site. In the special response element group, 10 *BnaZSNRT2s* might be involved in zein metabolism regulation (O2-site). In the tissue-specific expression element group, 9 *BnaZSNRT2s* might be related to meristem activation and expression (CCGTCC-box, CAT-box). In the biotic and abiotic-response *cis*-element group, 21, 11, and 10 *BnaZSNRT2s* might be involved in anoxia (ARE), low temperature (LTR), and defense and stress (TC-rich repeats) responsive processes. Moreover, we found that the predicted *cis*-acting elements in each group of the NRT2 gene family were similar *in B. napus* (Appendix A). In Groups II and III, 53% (9/17) and 70% (12/17) of the *cis*-acting elements existed in all members, respectively (Appendix A). In Group I, 31% (17/55) of the *cis*-acting elements were presented in the majority of candidates (Appendix A). These results indicate a complex regulatory network of *BnaZSNRT2s* responding to multi-factors and a relatively conserved regulation mechanism in each group, especially in Groups II and III.

In conclusion, our results suggested that *BnaZSNRT2s* may closely respond to plant hormones and abiotic stresses and may be regulated by many TF family members.

### 2.5. Potential miRNAs Targets of BnaZSNRT2s

We predicted the potential miRNA targets in the CDS sequences of the candidate 31 BnaZSNRT2s by psRNATarget online software (Appendix A, Figure 4). Accordingly, 21 miRNAs were found to have potential targets in 28 *BnaZSNRT2s*, except for *BnaZSNRT2.6C-5*, *BnaZSNRT2.6C-6,* and *BnaZSNRT2.8C-1*. In Group II, miR6030 might target both *BnaZSNRT2.5A-1* and *BnaZSNRT2.5C-1*; meanwhile, the latter might also be the target of miR395. In Group III, miR164 might target both *BnaZSNRT2.7A-1* and *BnaZSNRT2.7C-1*, and the latter might be the targets of miR166 and miR6031 as well. In Group I, a relatively complex regulation by multi-miRNAs was observed, where 16 BnaZSNRT2s might be targeted by 3 to 5 miRNAs, and miR397, miR164, miR172, miR395, and miR167 might target 10, 8, 8, 7, and 7 BnaZSNRT2s in this group, respectively. Moreover, single miRNA tends to target multiple BnaZSNRT2s in Group I−III, and the type of miRNA regulating each group was relatively conserved, indicating the relatively conserved expression and function in the same group. Additionally, miR164 might target 2 genes in Group III and 8 genes in Group I; miR395 might target 1 gene in Group II and 7 genes in Group I. The regulation of NRT2 homologs by miRNAs in Group I is more complicated than in Group II/III, implicating the diverse expression profile and function of Group I members.

### 2.6. Spatial-Temporal Expressions of BnaZSNRT2s in Different Developmental Stages of B. napus

To study the expression patterns of *BnaZSNRT2s* in extensive tissues/organs across distinct developmental stages in *B. napus*, a public RNA-seq dataset BnTIR (http://yanglab.hzau.edu.cn, accessed on 10 September 2020) including 52 samples was applied. Except for 14 genes with no detectable expression levels (TPM < 1) in all tissues investigated, the remaining 17 genes belonging to Group I−III have obvious preferential expression profiles in ‘ZS11’. The genes in the three groups generally have distinct expression profiles (Figure 5A). In general, the homologs in the same group had conserved expression patterns, implicating their potentially functional conservation and redundancy. In Group I, nearly all of the 13 genes were commonly highly expressed in roots (except *BnaZSNRT2.6A-4*) and showed uneven expression levels in shoot tissues with 4 genes (*BnaZSNRT2.6A-2*, *BnaZSNRT2.6C-2*, *BnaZSNRT2.6A-4,* and *BnaZSNRT2.6C-3*) having higher expression levels in the late developmental stages of silique and seed tissues as well. This implicated that the homologs in Group I might perform diverse functions in *B. napus*. In contrast, the expression profiles were highly conserved in the last two groups. In Group II, the 2 members (*BnaZSNRT2.5A-1* and *BnaZSNRT2.5C-1*) were highly expressed in leaf and silique tissues with a gradually increased in leaf (Figure 5B). Similarly, the 2 members of Group III (*BnaZSNRT2.7A-1* and *BnaZSNRT2.7C-1*) were highly expressed in leaf, silique, and seed tissues, and the expression levels were stable in leaf instead of silique and seed tissues (Figure 5B). The similar expression patterns between Group II and III indicated their close relationship and even similar functions. Given the general biological functions of the organs in the plant, we speculated that members of Group I may be related to the N uptake and transport in roots, while members of Group II and III may involve in N transport, storage, and/or accumulation in *B. napus*. Additionally, the Pearson correlation coefficient of 6 (75%) sister pairs was ≥0.8 (Appendix A), indicating their expression conservation and even functional redundancy.

### 2.7. Expression Profile of BnaZSNRT2s under LN Stress

To discover the potential functions of *BnaZSNRT2s* in the N utilization process in *B. napus*, an LN stress RNA-seq dataset (PRJNA612634) was applied in this study. Excepting 13 *BnaZSNRT2s* that were not expressed (FPKM < 1) in all samples, the last 18 expressed *BnaZSNRT2s* belong to three groups (Figure 6A). Consistent with the large number, up to 14 members in Group I had detectable transcript accumulation (FPKM ≥ 1) in the dataset, which was preferentially expressed in roots. Their expressions were significantly up-regulated under LN treatment after 3-, 5-, and 12-days (Figure 6B). Moreover, their expression patterns were somewhat different: *BnaZSNRT2.6A-2*, *BnaZSNRT2.4A-1*, *BnaZSNRT2.6C-2,* and *BnaZSNRT2.4C-1* were obviously preferentially up-regulated after 12 days LN treatment in roots; *BnaZSNRT2.1A-2*, *BnaZSNRT2.1A-3*, *BnaZSNRT2.1C-3,* and *BnaZSNRT2.8C-1* were significantly up-regulated after 5- and 12-days LN treatments; and the rest 6 members were up-regulated after 3-, 5- and 12-days LN treatment. This implicated the expression profile diversification of the homologs in Group I. Consistent with the spatial-temporal expression profile (Figure 5), the genes in Group II (*BnaZSNRT2.5A-1* and *BnaZSNRT2.5C-1*) and III (*BnaZSNRT2.7A-1* and *BnaZSNRT2.7C-1*) had highly conserved expression patterns under LN treatments respectively. The expressions of *BnaZSNRT2.7A-1* and *BnaZSNRT2.7C-1* in Group III were slightly up-regulated after 3-, 5- and 12-days LN treatment (Figure 6B). Whereas the expression levels of *BnaZSNRT2.5A-1* and *BnaZSNRT2.5C-1* in Group II were dramatically increased in roots after 12-days of LN treatment. Their expression levels were both peaked in leaf and roots after 12-day LN treatment, showing long-term N deficiency expression profiles. Notably, *BnaZSNRT2.5A-1* and *BnaZSNRT2.5C-1* showed very low expression levels in normal conditions in roots (Figure 5A), but their expressions were significantly up-regulated in roots under LN stress (Figure 6A), indicating their potential roles in roots in response to LN stress. To confirm the LN-induced expression profiles of *BnaZSNRT2.5A-1* and *BnaZSNRT2.5C-1* obtained from the RNA-seq dataset, an RT-qPCR assay was further applied. As shown in Figure 6C, the expression profiles of *BnaZSNRT2.5A-1* and *BnaZSNRT2.5C-1* were similar and were significantly up-regulated after long-term LN treatments, supporting the credibility of RNA-seq analysis. Notably, a peak expression level was observed at 7 days under LN treatment in the RT-qPCR assay, suggesting their expression trend under the LN stress condition.

## 3. Discussion

### 3.1. A Conserved Patterns of NRT2s Intron Insertion Patterns and Phylogeny

Numerous studies have indicated that the intron insertion patterns were commonly conserved in each gene family or subfamily in plants [39,40,41]. In this study, we identified 19 *BnaDarNRT2s* and 31 *BnaZSNRT2s* in two *B. napus* ecotypes. A total of 8 intron insertion sites were observed in *BnaDarNRT2s*, *BnaZSNRT2s,* and *AtNRT2s*, with “5” and “6” were nearly absolutely conserved in Group I, “4” and “7” were completely conserved in Group II, and “7” were conserved in Group III (Figure 1C). To our knowledge, this is the first time to focus on the intron insertion patterns of this family in plants. To further confirm our results in *Arabidopsis* and *B. napus*, we expand the analyses to 21 plant species, consisting of chlorophyte, bryophyte, basic angiosperm, monocots, and eudicots (Figure 7, Appendix A). We found that the intron patterns of this gene family generally tracked with accepted taxonomy across viridiplantae in an evolutionary context. For the 8 intron insertion sites, especially the highly conserved “4”, “5”, “6”, and “7” sites, none were distributed in chlorophyte. The intron “5” was first found in Bryophyte and was conserved in *P. patens* and *S. fallax* and investigated. Introns “5”, “6”, and “7” were observed as early as in basic angiosperm *A. trichopoda* and *C. kanehirae*; all of the 4 sites were observed in angiosperms while all of the investigated dicots genes had introns, and introns “4”, “5”, “6” and “7” were widely observed. However, only 24.1% (7/29) of monocot genes had introns with 5 genes sharing intron “7”. Notably, intron “4” was only distributed in Brassicaceae, indicating a new origin in this lineage. Overall, the intron insertion sites might be firstly derived after the divergence of chlorophyte and embryophyte, and intron “5” might be an ancestor and widely distributed in embryophyte except for monocots. An obvious bias trend between monocots and dicots was observed according to the divergence of intron sites.

To date, the NRT2 gene family has been identified in numerous plant genomes, such as *Brachypodium* [42], rice [43], barley [37], *M. esculenta* [44], and *A. thaliana* [35]. These studies generally classified this gene family into three groups, namely I−III. To gain insights into the evolutionary mechanisms and sequence features of the NRT2 gene family in the plant kingdom, we further identified the candidates in 22 plant genomes in Phytozome v13 ranging from chlorophyte to angiosperms, including *C. reinhardtii*, *V. carterigenomes*, *P. patens*, *S. fallax*, *A. trichopoda*, *C. kanehirae*, *A. coerulea*, *A. lyrata*, *A. thaliana*, *C. papaya*, *G. max*, *M. esculenta*, *M. guttatus*, *M. truncatula*, *P. persica*, *P. trichocarpa*, *R. communis*, *B. distachyon*, *O. sativa*, *S. bicolor*, *S. italica* and *Z. mays*. The *NRT2s* were observed in all of these species, with the number ranging from 2 (*A. trichopod* and *S. fallax*) to 16 (*M. guttatus*) (Appendix A). Consistent with previous studies [45,46], based on the systematic identification and phylogenetic analyses, we divided this gene family into five groups, including Group I (homologs of *AtNRT2.1*−*AtNRT2.4* and *AtNRT2.6*), Group II (homologs of *AtNRT2.5*), Group III (homologs of *AtNRT2.7*), chlorophyte-specific group (Group IV), and bryophyte-specific group (Group V) (Figure 7). Interestingly, we found that the intron distribution trend was highly consistent with the phylogenetic relationship of this gene family in viridiplantae: the genes with both introns “5” and “6” in angiosperms except monocots were observed in Group I. The genes with both introns “4” and “7” were clustered in Group II, and those with only intron “7” were in Group III. Notably, though the genes in dicots and monocots were commonly clustered into each of the angiosperm-specific groups (I−III), members of these two lineages shared the same intron pattern only in Group III, and the monocot genes in the last two groups were generally intron-less instead. Overall, our analyses indicated that the intron patterns might reflect the phylogeny of *NRT2s* even in an evolutionary context.

### 3.2. Expression and Function Characteristics of NRT2s in Plants

In this study, we revealed that the genes in Group I have a wide and less conserved expression pattern in *B. napus,* most of which were expressed in roots, whereas those in Groups II and III have a higher and conserved expression profile in the acrial part, especially in leaves and siliques (Figure 5). A similar situation was observed in many other eudicots [19,35,36,37,42,44,47,48]. For example, in Group I, many members were expressed abundantly in roots and a few other organs, such as *MtNRT2.1* in *M. truncatula* [47], *MeNRT2.1*−*MeNRT2.4* in *M. esculenta* [44], and *AtNRT2.1*, *AtNRT2.3*, *AtNRT2.4* and *AtNRT2.6* in *A. thaliana* [35]; In Group II, the genes generally showed strong transcripts in shoot tissues (e.g., stems and leaves) and weak expression levels in roots, such as poplar *PtNRT2.5A* and *PtNRT2.5B* [36], *M. truncatula MtNRT2.3* [47], and *M. esculenta MeNRT2.5* [44]; while members of Group III were specially expressed in shoots, such as *PtNRT2.7* [36] and *AtNRT2.7* [35]. Notably, the expression profiles of this gene family in monocots are different from that of dicots, especially those of Group II and III that showed higher expression levels in both roots and leaves (e.g., *HvNRT2.10* [37], *BdNRT2.5* in *B. distachyon* [42], *OsNRT2.3* in rice [48]; *BdNRT2.7* [42] and *OsNRT2.4* [48]), whereas the expression pattern of Group I was similar to eudicots, such as *HvNRT2.2*−*HvNRT2.9* in barley [37] and *OsNRT2.1*/*OsNRT2.2* in rice [48]. Together, the spatial-temporal expression profile of *NRT2s* in Group I was generally similar between monocots and eudicots, whereas those of Group II and III were distinct.

Consistent with their critical roles in nitrate transporters, the *NRT2s* generally responded positively to external nitrate or N deficiency in numerous plants. For example, *AtNRT2.1*/*AtNRT2.4*/*AtNRT2.6*, *MeNRT2.1*−*2.4*, *HvNRT2.2*−*2.9* and *OsNRT2.1*−*2.2* in Group I, and *AtNRT2.5*, *MeNRT2.5*, *HvNRT2.1,* and *OsNRT2.3* in Group II were up-regulated under low nitrate or LN treatment in roots [35,37,43,44]. At the same time, *AtNRT2.7* in Group III was intensively induced by limited nitrate supply in shoots [35]. Similarly, we found that the *BnaZSNRT2s* in Group I and II were significantly up-regulated in roots under LN treatment, whereas these in Group III were positively induced in leaves under LN treatment (Figure 6). Notably, in contrast to the situation in Group I and III, the genes in Group II showed an obvious inverse trend between the spatial-temporal (Figure 5) and LN stress (Figure 6) expression profiles which were up-regulated in roots under LN treatments instead of leaves. Moreover, the LN stress expression profile of this gene family is generally consistent with their known functions in plants. For instance, members of Group I were widely proved to function in roots under nitrate or N starvation, e.g., *MeNRT2.2* [44], *CmNRT2.1* [49], and *AtNRT2.1* [50]. *OsNRT2.3a* in Group II acted as a long-distance nitrate transport from roots to shoots under low nitrate treatment [51], while members of Group III mainly functioned in seeds, such as wheat *TaNRT2.5* [52] and *AtNRT2.7* [24]. These results suggested a spatial-temporal collaborative role of this gene family in the root and shoot tissues in N utilization in plants.

## 4. Materials and Methods

### 4.1. Identification of NRT2 Genes in Plants

The *AtNRT2s* were acquired from the TAIR database (http://www.arabidopsis.org, accessed on 11 July 2019). To identify the *NRT2s* in the *B. napus* genome, BLASTP and Tblastn searches were performed in GENOSCOPE (http://www.genoscope.cns.fr/brassicanapus/, accessed on 19 August 2014) and BnPIR (http://cbi.hzau.edu.cn/bnapus/, accessed on 10 September 2020) databases respectively, using the *AtNRT2s* as the queries with a low-stringency criterion (cutoff *p* < 0.1). After deleting the redundant sequences, the remainders were confirmed by HMMER (https://www.ebi.ac.uk/Tools/hmmer/, accessed on 19 November 2021) and SMART (http://smart.embl-heidelberg.de/, accessed on 26 October 2020) tools to ensure they had the characteristic domains of the NRT2 family. DNA, cDNA, and encoding protein sequences of the candidate genes were acquired from the GENOSCOPE and BnPIR database. We also identified the *NRT2s* in *C. reinhardtii*, *V. carterigenomes*, *P. patens*, *S. fallax*, *A. trichopoda*, *C. kanehirae*, *B. rapa*, *B. oleracea*, *A. coerulea*, *A. lyrata*, *C. papaya*, *G. max*, *M. esculenta*, *M. guttatus*, *M. truncatula*, *P. persica*, *P. trichocarpa*, *R. communis*, *B. distachyon*, *O. sativa*, *S. bicolor*, *S. italica* and *Z. mays* from Phytozome v13 database (https://phytozome-next.jgi.doe.gov/, accessed on 22 November 2011) [53] by the same method. The physicochemical properties and subcellular localization analyses of candidates were predicted by ExPASy online tool (http://www.expasy.org/tools/, accessed on 23 February 2022) [54] and by Cell-PLoc-2 (http://www.csbio.sjtu.edu.cn/bioinf/Cell-PLoc-2/, accessed on 28 June 2010) and WoLF PSORT (https://www.genscript.com/wolf-psort.html, accessed on 21 May 2007), respectively.

### 4.2. Phylogenetic and Sequence Structure Analysis of B. napus NRT2 Gene Family

To discover the evolutionary relationship of the NRT2 family in *B. napus* and *Arabidopsis*, the protein sequences of candidates in *B. napus* (*BnaDarNRT2s* and *BnaZSNRT2s*) and *Arabidopsis* (*AtNRT2s*) were applied to perform a multiple sequence alignment analyses by MAFFT version 7 tool with default parameters (https://mafft.cbrc.jp/alignment/server/, accessed on 06 September 2017). Based on multiple sequence alignment analyses, a phylogenetic tree was generated by MEGA version 7 [55] using the NJ method with the following major parameters: Poisson correction, bootstrap with 1000 replicates, and pairwise deletion. The sequence structures of *BnaDarNRT2s*, *BnaZSNRT2s,* and *AtNRT2s* were analyzed by a Gene Structure Display Server (GSDS) 2.0 (http://gsds.cbi.pku.edu.cn/, accessed on 10 December 2014) [56], using the cDNA and protein sequences of candidates. The domains and TM regions of BnaDarNRT2s, BnaZSNRT2s, and AtNRT2s were predicted by the HMMER tool (https://www.ebi.ac.uk/Tools/hmmer/, accessed on 19 November 2021), and then were visualized by Weblogo online software (http://weblogo.threeplusone.com/, accessed on 06 June 2004). The intron insertion sites and phases of the candidates were obtained by comparing the DNA and cDNA sequences of each gene manually using MEGA version 7.

### 4.3. Chromosomal Location and Collinearity Relationship of BnaZSNRT2s

The chromosome location information of candidate *BnaZSNRT2s* was acquired from the BnPIR database. The chromosome map of candidates was drawn by MapChart v2.32 software. The collinearity relationship of *BnaZSNRT2s*, *BrNRT2s*, *BoNRT2s*, and *AtNRT2s* was calculated by the CoGe online tool (https://genomevolution.org/coge/, accessed on 04 February 2008). The duplication events of *BnaZSNRT2s* were identified in the previous study [57].

### 4.4. The Potential Regulatory Mechanism in the Promoter Regions of BnaZSNRT2s

The −1500 bp upstream promoter sequence of *BnaZSNRT2s* was used to predict the potential TFs regulation information by the PlantTFDB database (http://planttfdb.gao-lab.org/, accessed on 24 October 2016) with a threshold *p*-value < 10^−6^. The potential *cis*-elements in the promoter regions of *BnaZSNRT2s* (−1500 bp upstream sequence) were predicted by PlantCARE online software (http://bioinformatics.psb.ugent.be/webtools/plantcare/html/, accessed on 11 September 2000). The regulation network of *BnaZSNRT2s* was generated by Cytoscape 3.8.2 software [58]. The potential miRNAs regulating sites of *BnaZSNRT2s* were analyzed by the psRNATarget website (expectation ≤ 6) (http://plantgrn.noble.org/psRNATarget/, accessed on 30 April 2018).

### 4.5. Spatial-Temporal and LN Stress Expression Profile Analysis of BnaZSNRT2s by RNA-Seq Data

To explore the temporal-spatial expression patterns of *BnaZSNRT2s* at different developmental stages in *B. napus*, the RNA-Seq dataset in BnTIR (http://yanglab.hzau.edu.cn/, accessed on 10 September 2020) was obtained. The heatmap was generated by Cluster 3.0 [59] and Java Treeview software [60] according to log2-transformed data. The genes with FPKM < 1 were speculated to be pseudogenes or specifically expressed genes under other conditions, and thus were not included in the heatmap in this study. Similarly, to explore the LN-inductive expression patterns of *BnaZSNRT2s*, an LN stress RNA-Seq dataset of ‘ZS11’ ecotype seedling roots and leave was acquired from NCBI (BioProject: PRJNA612634). The heatmap was generated by Cluster 3.0 [59] and Java Treeview software [60] based on the log2-transformed data of candidates.

### 4.6. RT-qPCR Analysis of BnaZSNRT2s under LN Conditions

Seeds of ZS11 were acquired from the College of Agriculture and Biotechnology, Southwest University. The treatments of the seeds and seedlings referred to a previous study [61]. The normal nutrient solution comprised of 0.5 mM K_2_SO_4_, 0.25 mM KH_2_PO_4_, 325 µM MgSO_4_, 50 µM NaCl, 8 µM H_3_BO_3_, 0.4 µM MnSO_4_, 0.4 µM ZnSO_4_, 0.4 µM CuSO_4_, 0.1 µM Na_2_MoO_4_, 40 µM Fe-EDDHA, 10 µM C_2_H_4_N_4_, 1.8 mM Ca(NO_3_)_2_ and 0.2 µM (NH_4_)_2_SO_4_ (control, CK). For the LN treatment, 1.8 mM Ca(NO_3_)_2_ and 0.2 mM (NH_4_)_2_SO_4_ were replaced by 0.09 mM Ca(NO_3_)_2_ and 0.001 mM (NH_4_)_2_SO_4_ [62]. The pH of the solution was 5.8. The root tissues were reaped at 1, 3, 5, 7, and 12 days after the treatments, which were promptly frozen in liquid nitrogen, and then stored at −80 °C for the purpose of RNA isolation. The RT-qPCR analysis method referred to our previous study [61], using the GoScriptTM Reverse Transcription Mix, Oligo(dT) kit (Promega, Beijing, China), and Taq Pro Universal SYBR qPCR Master Mix (Vazyme, Nanjing, China). The primers were designed by Primer Premier 5 software, and the primer sequences were presented in Appendix A. Ultimately, we acquired the data (mean ± standard deviation) of all three independent repeated trials and calculated the relative expression of *BnaZSNRT2s* by the 2(−∆∆Ct) method. Standard errors were presented by Error bars from three independent repeated trials. Expression level Differences of *BnaZSNRT2s* were assessed by a One-way ANOVA test (* *p* < 0.05; ** *p* < 0.01) using Excel 2016.

## Figures and Tables

**Figure 1 ijms-23-04965-f001:**
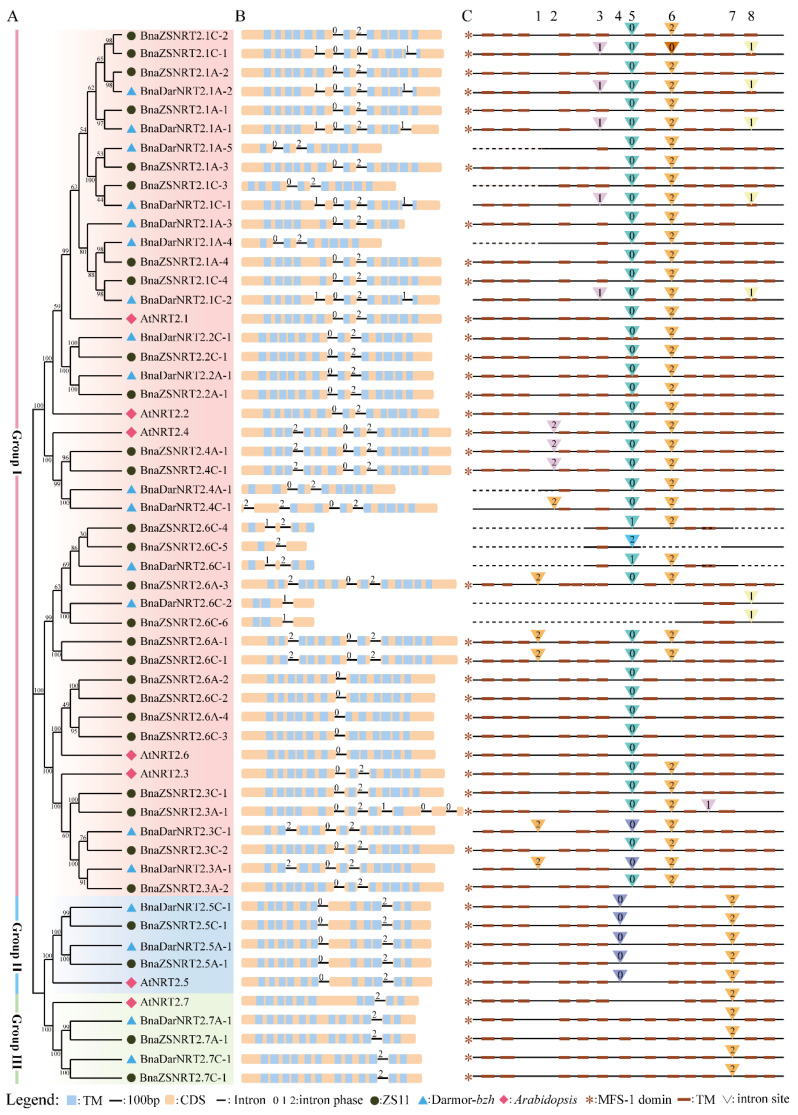
Phylogenetic tree and sequence structures of the candidate *NRT2s* in *B. napus* and *Arabidopsis*. (**A**) Neighbor-joining (NJ) phylogenetic tree of *NRT2s* in *B. napus* and *Arabidopsis*. Different background colors represent different groups. (**B**) Gene structures of *NRT2s* in *B. napus* and *Arabidopsis*. Exons are shown by yellow boxes, transmembrane (TM) domains are shown by blue boxes, and the lines between the colored boxes correspond to the introns. Numbers 0, 1, and 2 represent introns phase 0, 1, and 2, respectively. (**C**) Intron insertion patterns of *NRT2s* in *B. napus* and *Arabidopsis*. The top numbers (1–8) indicate the orders of the 8 conserved intron sites. The triangles represent different intron insertion sites. Each column (triangle with the same color and number) represents the same intron insertion site and phase. The dashed lines represent the severely missing sequences of candidates.

**Figure 2 ijms-23-04965-f002:**
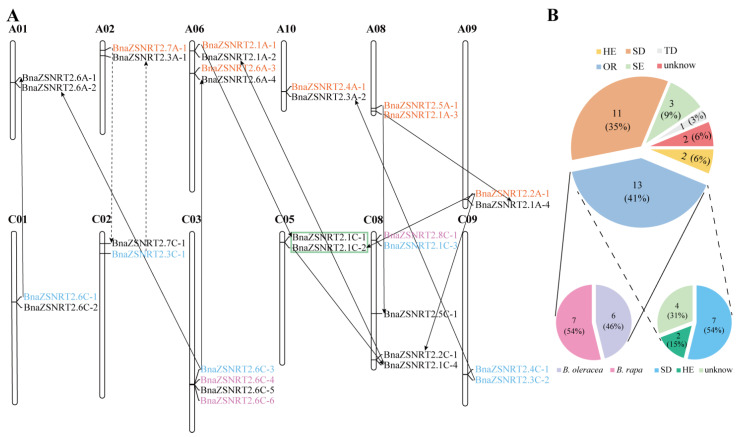
Chromosome distribution and duplication of *BnaZSNRT2s*. (**A**) 31 *BnaZSNRT2s* were mapped on 12 chromosomes in *B. napus*. The genes in orange and blue originated from *B. rapa* and *B. oleracea,* respectively; genes in purple were involved in segmental exchange (SE) events; the genes with green frames were tandem duplication (TD) pairs. The black and dashed lines with an arrow represent the duplication direction of genes involved in segmental duplication (SD) and homologous exchange (HE) events, respectively. (**B**) The big pie chart represents the percentage of *BnaZSNRT2s* derived from SD, SE, HE, TD, and orthologous region (OR) events, respectively; the left small pie chart represents the percentage of *BnaZSNRT2s* from OR and then experienced small duplication events in *B. napus*, and the right small pie chart represents the percentage of *BnaZSNRT2s* involved in OR events from *B. rapa* and *B. oleracea*.

**Figure 3 ijms-23-04965-f003:**
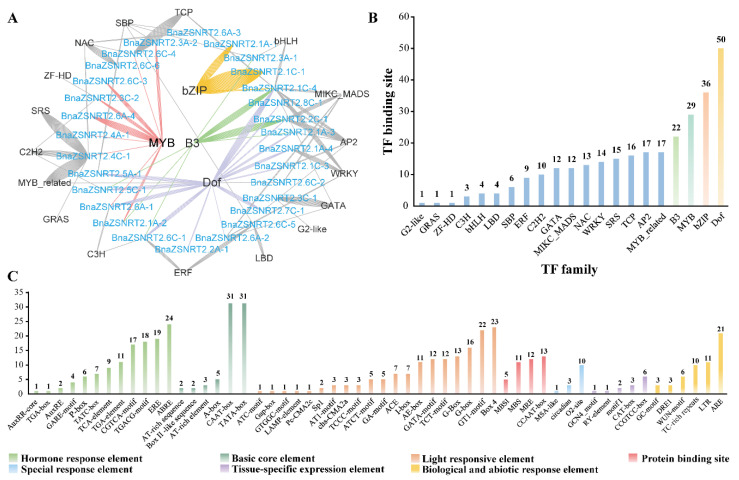
Transcription factor (TF) binding network and *cis*-element analysis in the promoter regions of *BnaZSNRT2s*. (**A**) The potential TF binding network of the 31 *BnaZSNRT2s* predicted by the PlantTFDB tool. (**B**) The TF gene families with potential binding sites in the promoter regions of the *BnaZSNRT2s*. (**C**) The *cis*-elements in the promoter regions of the *BnaZSNRT2s*. The ordinate represents the number of *BnaZSNRT2s*.

**Figure 4 ijms-23-04965-f004:**
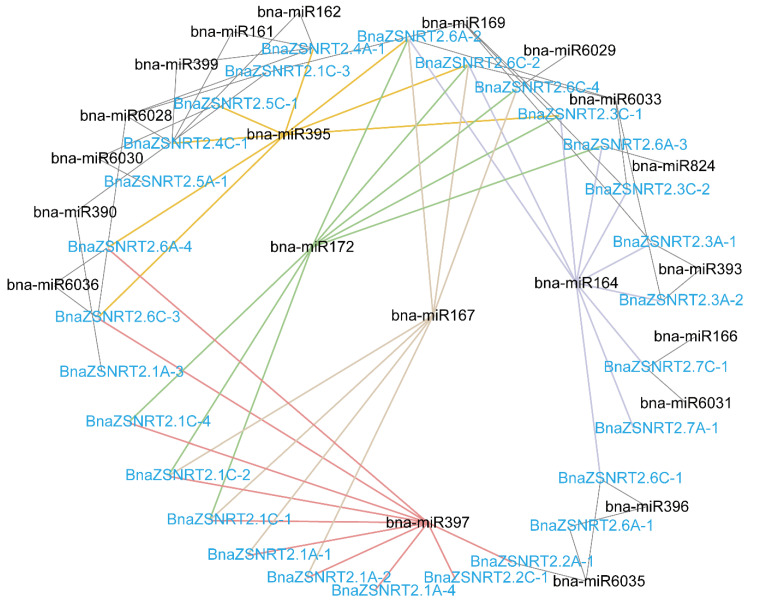
The potential miRNA targeting network of the *BnaZSNRT2s*. A total of 21 miRNAs with black were predicted. *BnaZSNRT2s* were represented in blue.

**Figure 5 ijms-23-04965-f005:**
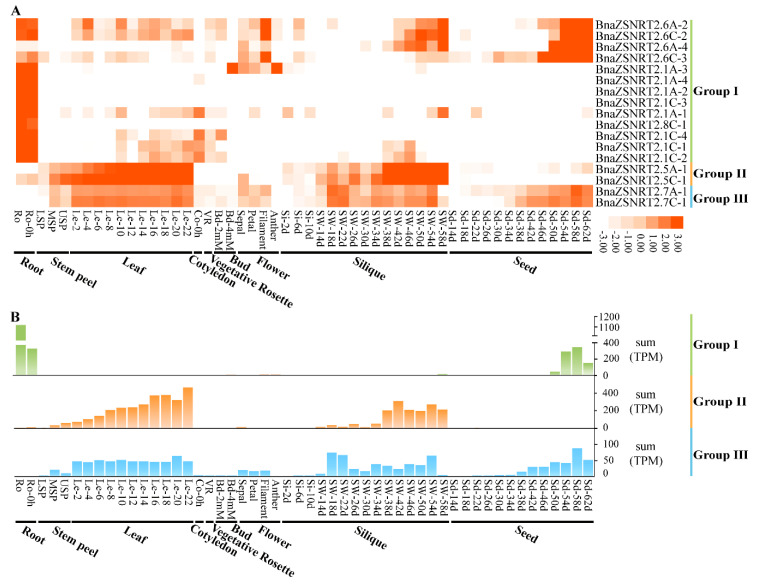
Expression pattern of *BnaZSNRT2s* in *B. napus* at different developmental stages. (**A**) The expression profile of *BnaZSNRT2s* in 52 tissues. (**B**) The sum of expression levels in Group I−III. The ordinate represents the sum of expression levels (TPM). “Ro” = root, “LSP” = lower stem peel, “MSP” = middle stem peel, “USP” = upper stem peel, “Le” = leaf, “Co” = cotyledon, “VR” = vegetative rosette, “Bd” = bud, “Si” = silique, “SW” = silique wall, “Sd” = seed; “h,” and “d,” indicate hour and day, respectively.

**Figure 6 ijms-23-04965-f006:**
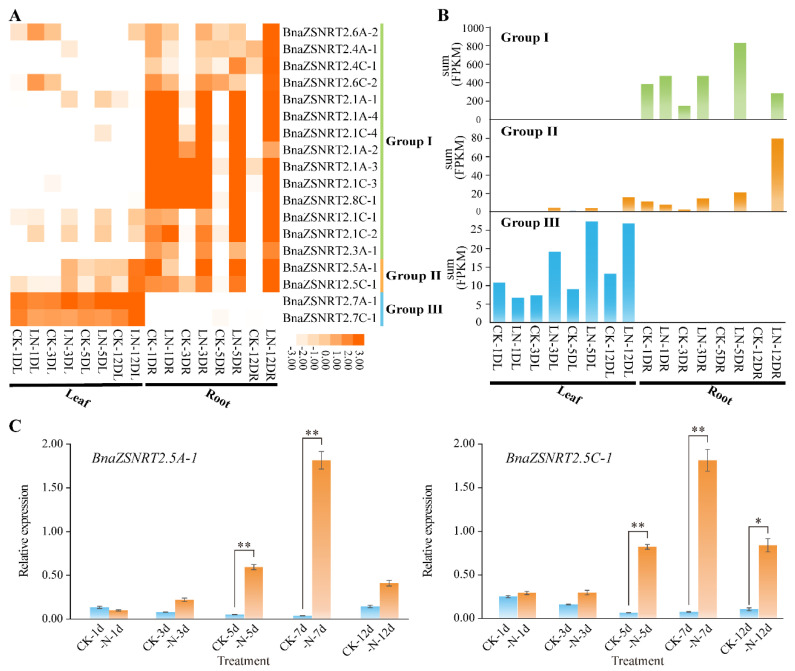
Expression pattern of *BnaZSNRT2s* under low nitrogen (LN) stress treatments. (**A**) The expression profiles of *BnaZSNRT2s* in ‘ZS11’ seedling leaf and root tissues under LN stress treatment based on RNA-Seq dataset. “CK” = normal N condition, “LN” = Low N; “D”, “L”, and “R” indicate day, leaf, and root, respectively. (**B**) The sum of expression levels in each group. The ordinate represents the sum of expression levels (FPKM) corresponding to each tissue investigated. (**C**) Expression levels of *BnaZSNRT2.5A-1* and *BnaZSNRT2.5C-1* under LN treatments by RT-qPCR method. The reference genes were *BnaActin7* (GenBank accession no. AF024716) and *BnaUBI* (GenBank accession no. NC027770). Error bars represent the standard deviation of three independent experiments. *: Significant difference (0.05 > *p* > 0.01); **: Extremely significant difference.

**Figure 7 ijms-23-04965-f007:**
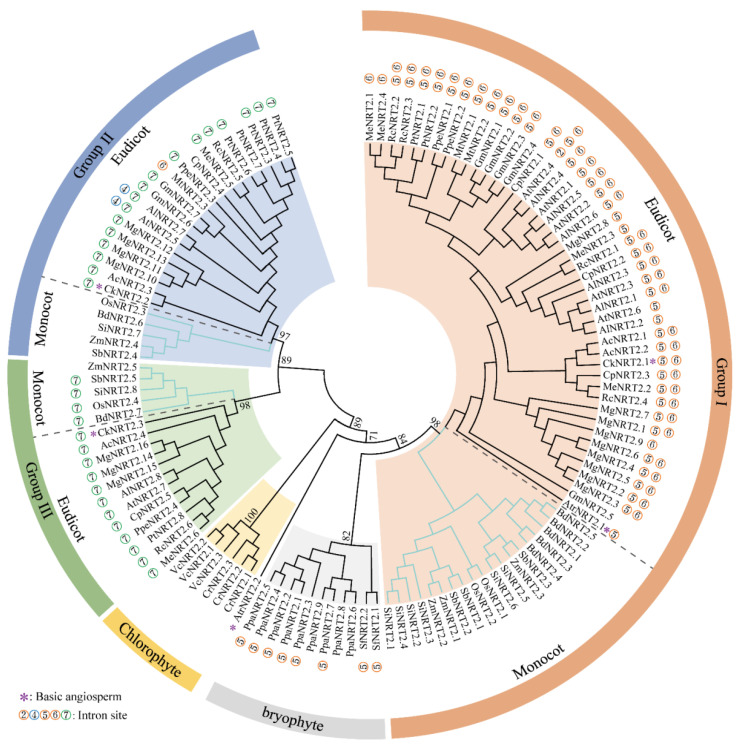
Phylogenetic analysis of NRT2 gene families in 22 plants. Different groups were colored by a special background. Monocot and eudicot within Groups I–III were divided by dashed lines. The number beside the branch indicates the bootstrap value of each group from 1000 replicates.

**Table 1 ijms-23-04965-t001:** Features of the 31 NRT2 genes (BnaZSNRT2s) identified in Brassica napus Zhongshuang 11 (ZS11) ecotype.

Gene Name	Genome ID	Chromosome	Protein Length (aa)	CDS Length (bp)	DNA Length (bp)	pI	Molecular Weight (kDa)	Subcellular Localization	
								Predicted by Cell-PLoc2.0	Predicted by WoLF PSORT
*BnaZSNRT2.6A-1*	BnaA01G0234100ZS	chrA01	546	1641	2624	7.23	59.39	Cell membrane	plasmalemma
*BnaZSNRT2.6A-2*	BnaA01G0234200ZS	chrA01	541	1626	1721	8.80	58.48	Cell membrane	plasmalemma
*BnaZSNRT2.7A-1*	BnaA02G0054200ZS	chrA02	484	1455	1525	8.14	52.06	Cell membrane	plasmalemma
*BnaZSNRT2.3A-1*	BnaA02G0096600ZS	chrA02	502	1509	6544	9.06	53.76	Cell membrane	plasmalemma
*BnaZSNRT2.1A-1*	BnaA06G0047500ZS	chrA06	530	1593	2163	9.07	57.70	Cell membrane	plasmalemma
*BnaZSNRT2.1A-2*	BnaA06G0047600ZS	chrA06	530	1593	1804	9.03	57.73	Cell membrane	plasmalemma
*BnaZSNRT2.6A-3*	BnaA06G0186600ZS	chrA06	543	1632	2940	6.89	59.12	Cell membrane	plasmalemma
*BnaZSNRT2.6A-4*	BnaA06G0186700ZS	chrA06	538	1617	1720	9.04	58.37	Cell membrane	plasmalemma
*BnaZSNRT2.5A-1*	BnaA08G0276500ZS	chrA08	499	1500	1773	9.01	54.17	Cell membrane	plasmalemma
*BnaZSNRT2.1A-3*	BnaA08G0300800ZS	chrA08	530	1593	2129	8.79	57.61	Cell membrane	plasmalemma
*BnaZSNRT2.2A-1*	BnaA09G0667700ZS	chrA09	506	1521	1983	9.02	54.83	Cell membrane	plasmalemma
*BnaZSNRT2.1A-4*	BnaA09G0667800ZS	chrA09	529	1590	1860	8.90	57.34	Cell membrane	plasmalemma
*BnaZSNRT2.4A-1*	BnaA10G0160100ZS	chrA10	527	1584	2368	8.90	57.61	Cell membrane	plasmalemma
*BnaZSNRT2.3A-2*	BnaA10G0160300ZS	chrA10	536	1611	1964	9.14	58.21	Cell membrane	plasmalemma
*BnaZSNRT2.6C-1*	BnaC01G0301400ZS	chrC01	546	1641	2618	7.66	59.40	Cell membrane	plasmalemma
*BnaZSNRT2.6C-2*	BnaC01G0301600ZS	chrC01	541	1626	1721	8.85	58.56	Cell membrane	plasmalemma
*BnaZSNRT2.7C-1*	BnaC02G0063100ZS	chrC02	502	1509	1579	7.61	53.45	Cell membrane	plasmalemma
*BnaZSNRT2.3C-1*	BnaC02G0116100ZS	chrC02	536	1611	2105	9.04	58.06	Cell membrane	plasmalemma
*BnaZSNRT2.6C-3*	BnaC03G0602800ZS	chrC03	538	1617	1720	9.13	58.13	Cell membrane	plasmalemma
*BnaZSNRT2.6C-4*	BnaC03G0603000ZS	chrC03	154	465	1109	9.86	17.09	Cell membrane	cytoplasm
*BnaZSNRT2.6C-5*	BnaC03G0603300ZS	chrC03	162	489	17582	6.91	17.55	Chlo Cyto	E.R.
*BnaZSNRT2.6C-6*	BnaC03G0603600ZS	chrC03	184	555	652	9.93	19.86	Cell membrane	chloroplast
*BnaZSNRT2.1C-1*	BnaC05G0059500ZS	chrC05	475	1428	1806	9.03	51.83	Cell membrane	plasmalemma
*BnaZSNRT2.1C-2*	BnaC05G0059600ZS	chrC05	530	1593	2059	9.03	57.76	Cell membrane	plasmalemma
*BnaZSNRT2.8C-1*	BnaC08G0033200ZS	chrC08	113	342	342	6.02	12.19	Cell membrane	cytoplasm
*BnaZSNRT2.1C-3*	BnaC08G0033300ZS	chrC08	395	1188	1579	9.00	43.15	Cell membrane	chloroplast
*BnaZSNRT2.5C-1*	BnaC08G0220400ZS	chrC08	498	1497	1799	9.11	54.13	Cell membrane	plasmalemma
*BnaZSNRT2.2C-1*	BnaC08G0532700ZS	chrC08	502	1509	1967	8.91	54.40	Cell membrane	plasmalemma
*BnaZSNRT2.1C-4*	BnaC08G0532800ZS	chrC08	529	1590	1915	8.95	57.38	Cell membrane	plasmalemma
*BnaZSNRT2.4C-1*	BnaC09G0443000ZS	chrC09	527	1584	4336	8.90	57.52	Cell membrane	plasmalemma
*BnaZSNRT2.3C-2*	BnaC09G0443100ZS	chrC09	567	1704	1987	9.08	61.87	Cell membrane	plasmalemma

Abbreviations: aa, amino acids; CDS, coding sequence; pI, isoelectric point; E.R., endoplasmic reticulum; Chlo, chloroplast; Cyto, Cytoplasm.

## Data Availability

The study did not report any data.

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
