# Peer review of "Genome-Wide Characterization of High-Affinity Nitrate Transporter 2 (NRT2) Gene Family in Brassica napus"

_ijms, 2022, doi:10.3390/ijms23094965_

Round 1

Reviewer 1 Report

The present study aims to identify all possible candidate NRT2s genes from two B. napus genomes. The authors identified 19 and 31 candidate BnaNRT2s genes in ‘Darmor-bzh’ (BnaDarNRT2s) and ‘Zhongshuang11′ (BnaZSNRT2s), respectively. They also finished the analysis of phylogentic trees and possible expression pattern in all candidate NRT2s gene family. The GWAS analysis, clustering, and bioinformatics analysis are logically and reasonable. The English writing is well. Only minor concerns need to clarify, as follow:

  1. The authors need to describe the nomination rule of BnaNRT2s from two different ecotypes. According to the Table and supplementary tables, they nominated the sequences following the located chromosomes in Genome A then Genome C in B. napus ‘Zhongshuang11′. Why? 1 (Genome A) is highly similar to BnaZSNRT2.15 (Genome C), the others have similar tendency. Are the belong to two different candidate genes or they are the same candidate gene and just belong to two alleles?
  2. Another concern, the same group candidate genes have quite similar expression patterns in Figure 5, Figure 6. Have common regulatory elements or domain in the same group?
  3. The format of cited references in the texture should be unified. Please check all the references.

Author Response

Dear Reviewer #1,

Thank you very much for your kindly suggestions. We have revised our manuscript according to your comments, carefully. In the following pages are our point-by-point responses to each comment of you. We use blue highlight for the main amends in the manuscript. We hope that our revisions in the manuscript and our accompanying responses will be sufficient to make our manuscript suitable for publication in your journal.

Thank you very much for your hard work in reviewing our manuscript.

Yours sincerely,

Hai Du

Reviewer #1

Comments and Suggestions for Authors

The present study aims to identify all possible candidate NRT2s genes from two B. napus genomes. The authors identified 19 and 31 candidate BnaNRT2s genes in ‘Darmor-bzh’ (BnaDarNRT2s) and ‘Zhongshuang11′ (BnaZSNRT2s), respectively. They also finished the analysis of phylogentic trees and possible expression pattern in all candidate NRT2s gene family. The GWAS analysis, clustering, and bioinformatics analysis are logically and reasonable. The English writing is well. Only minor concerns need to clarify, as follow:

1. The authors need to describe the nomination rule of BnaNRT2s from two different ecotypes. According to the Table and supplementary tables, they nominated the sequences following the located chromosomes in Genome A then Genome C in B. napus ‘Zhongshuang11′. Why? 1 (Genome A) is highly similar to BnaZSNRT2.15 (Genome C), the others have similar tendency. Are they belong to two different candidate genes or they are the same candidate gene and just belong to two alleles?

Response: Thank you for your suggestion. According to you and the Reviewer #2’s requirements, we have renamed the candidate genes based on A. thaliana NRT2 homologous genes and have added the chromosome (“A” and “C”) information in the new names to clarify the nomination rule clearly. As result of it, we have corrected/unify the gene names in the full text, and all of the figures and tables in the manuscript if it was necessary. Thank you.

Moreover, the genes in each gene pair are two candidate genes that are the homologs of a given Arabiddopsis NRT2 gene based on collinearity analysis. The colinearity gene pairs are not exactly equivalent to alleles, as the some of them were derived from segmental duplication event etc. In addition, we have compared the sequence identity and similarity of the full-length DNA, CDS and protein sequences of each gene pair in B. napus (Table S5). The result showed that the gene pair has high sequence similarity, such as BnaZSNRT2.1A-1/BnaZSNRT2.1C-1 (new name). These contents were added in section 2.3 in page 8. Thank you again.

2. Another concern, the same group candidate genes have quite similar expression patterns in Figure 5, Figure 6. Have common regulatory elements or domain in the same group?

Response: Thank you for your insightful comment. We have done the analyses about regulatory elements or domain. And the results showed that common regulatory elements or domain indeed existed in the same group. Accordingly, we have added the relevant contents in sections 2.2, 2.4 and 2.5, and have supplied a figure (Figure S1) to visualize the results of protein domains in this gene family. Thank you.

3. The format of cited references in the texture should be unified. Please check all the references.

Response: We have checked all the references in our revised manuscript and have corrected it if it was necessary. Thank you again.

Reviewer 2 Report

Nitrogen is a crucial element for plant development. NITRATE TRANSPORTER 2 (NRT2) homologs play key roles in nitrate uptake and even utilization in plants. Therefore, gene family analysis of NRT2 in Brassica napus may help in the molecular breeding selection of the crop.

The authors identified 31 and 17 NRT2  genes in B.napus ecotypes 'ZS11' and 'Darmor-bzh' respectively and analyzed their structure,  gene duplication and gene loss status, and expression patterns. They predicted the elements in promoters, and also get candidate siRNAs targeted to some NRT2 genes. Most methods seemed fine. 

Here are some comments:

  1. Line 91: Just blastp may not be enough for duplicate and gene loss analysis. The author could use tbastn to find the pseudogenes in chromosome sequences to analyze the gene loss.
  2. Line 101: If the authors have more time, they could consider adding 'A', 'C', or chromosome number in the gene name. or number based on A. thaliana NRT2s number. This type of name may be more clear to readers.
  3. Line 106-107, the author already had results to answer the question in section 2.2 (2.3 in the manuscript?).  Why not move section 2.2 to this part.
  4. Line 109-113. could add the average value and SD of these numbers.
  5. Table1. please consider using gene ID rather than transcript ID in column 2. (BnaA01G0234100ZS, not BnaA01T0234100ZS).
  6. Table S2. does sheet2 mean something?
  7. Line 179-181. consider using tblastn to find ancient gene sequences in genome DNA. 
  8. Line 195-197. is that accurate to use WGD in hybrid?  
  9. Figure2A. An important result, please improve the quality of the diagram. Try to reduce the number of crossed arrows and make it clear where the arrows are pointing.
  10. Line 287-289. To my knowledge, BnTIR is using TPM as the expression value, not FPKM, please double check it.
  11. Line 316-318. Can't find the RNA-seq dataset (PRJNA612634) in NCBI. Is it made by the authors? if so, could you provide the reviewer link, and add RNA-seq methods.

Thanks.

Author Response

Dear Reviewer #2,

Thank you very much for reviewing our manuscript. The comments of you were highly insightful and enabled us to improve the quality of our manuscript. We have tried our best to address all of your requirements. And we have revised our manuscript based on your comments. In the following pages are our point-by-point responses to each comment of you. We use blue highlight for the main amends in the manuscript. We hope that our revisions in the manuscript and our accompanying responses will be sufficient to make our manuscript suitable for publication in your journal.

Thank you very much for your hard work in reviewing our manuscript.

Yours sincerely,

Hai Du

Reviewer #2

Comments and Suggestions for Authors

Nitrogen is a crucial element for plant development. NITRATE TRANSPORTER 2 (NRT2) homologs play key roles in nitrate uptake and even utilization in plants. Therefore, gene family analysis of NRT2 in Brassica napus may help in the molecular breeding selection of the crop.

The authors identified 31 and 17 NRT2  genes in B.napus ecotypes 'ZS11' and 'Darmor-bzh' respectively and analyzed their structure,  gene duplication and gene loss status, and expression patterns. They predicted the elements in promoters, and also get candidate siRNAs targeted to some NRT2 genes. Most methods seemed fine. 

Here are some comments:

1. Line 91: Just blastp may not be enough for duplicate and gene loss analysis. The author could use tbastn to find the pseudogenes in chromosome sequences to analyze the gene loss.

Response: You are right. We have done the tblastn to verify our sequences . Thank you.

2. Line 101: If the authors have more time, they could consider adding 'A', 'C', or chromosome number in the gene name. or number based on A. thaliana NRT2s number. This type of name may be more clear to readers.

Response: According to your suggestion, we have renamed our candidates based on A. thaliana homologous NRT2s, and have added the chromosome (‘A’ and ‘C’ subgenomes) information in the new name as well. As result of it, we have corrected/unify the gene names in the full text, and all of the figures and tables in the manuscript if it was necessary. Please check them. Thank you.

3. Line 106-107, the author already had results to answer the question in section 2.2 (2.3 in the manuscript?).  Why not move section 2.2 to this part.

Response: Thank you for your comment. We have moved this content to section 2.3. Thank you.

4. Line 109-113. could add the average value and SD of these numbers.

Response: We have added the average value and SD information in line 109-113 and Table S1. Thank you.

5. please consider using gene ID rather than transcript ID in column 2. (BnaA01G0234100ZS, not BnaA01T0234100ZS).

Response: We have already amended it in our manuscript. Thank you.

6. Table S2. does sheet2 mean something?

Response: It was meaningless. We are sorry for this mistake. We have deleted it from the table. Thank you.

7. Line 179-181. consider using tblastn to find ancient gene sequences in genome DNA. 

Response: According to your suggestion, we have verified the data in our manuscript by tblastn method. And the results are consistent with the previous results. Thank you.

8. Line 195-197. is that accurate to use WGD in hybrid?

Response: Yes, the statement is not right. It would be the hybridization event between Brassica rapa and Brassica oleracea. We have replaced “WGD” with “allopolyploidy between B. rapa and B. oleracea” in line 195-197 in the manuscript. Thank you.

9. An important result, please improve the quality of the diagram. Try to reduce the number of crossed arrows and make it clear where the arrows are pointing.

Response: According to your requirement, we have redrawn the Figure 2A to make it clearly. Please check it. Thank you.

10. Line 287-289. To my knowledge, BnTIR is using TPM as the expression value, not FPKM, please double check it.

Response: We are sorry for this mistake. We have corrected it in our revised manuscript. Thank you.

11. Line 316-318. Can't find the RNA-seq dataset (PRJNA612634) in NCBI. Is it made by the authors? if so, could you provide the reviewer link, and add RNA-seq methods.

Response: Yes, this dataset was made by our lab. Currently, the dataset had been submitted to NCBI (ID: PRJNA612634). However, as we are preparing the manuscript based on the dataset, thus it is not public (it has not been released yet) in NCBI now. Thank you again.
